# Coagulopathies after Vaccination against SARS-CoV-2 May Be Derived from a Combined Effect of SARS-CoV-2 Spike Protein and Adenovirus Vector-Triggered Signaling Pathways

**DOI:** 10.3390/ijms221910791

**Published:** 2021-10-06

**Authors:** Ralf Kircheis

**Affiliations:** Syntacoll GmbH, 93342 Saal an der Donau, Germany; rkircheis@syntacoll.de; Tel.: +49-151-167-90606

**Keywords:** SARS-CoV-2, spike protein, thrombosis, thrombocytopenia, vaccine-induced immune thrombotic thrombocytopenia (VITT), NF-kappaB, adenoviral vector, platelet factor 4 (PF4), vaccination, COVID-19

## Abstract

Novel coronavirus SARS-CoV-2 has resulted in a global pandemic with worldwide 6-digit infection rates and thousands of death tolls daily. Enormous efforts are undertaken to achieve high coverage of immunization to reach herd immunity in order to stop the spread of SARS-CoV-2 infection. Several SARS-CoV-2 vaccines based on mRNA, viral vectors, or inactivated SARS-CoV-2 virus have been approved and are being applied worldwide. However, the recent increased numbers of normally very rare types of thromboses associated with thrombocytopenia have been reported, particularly in the context of the adenoviral vector vaccine ChAdOx1 nCoV-19 from Astra Zeneca. The statistical prevalence of these side effects seems to correlate with this particular vaccine type, i.e., adenoviral vector-based vaccines, but the exact molecular mechanisms are still not clear. The present review summarizes current data and hypotheses for molecular and cellular mechanisms into one integrated hypothesis indicating that coagulopathies, including thromboses, thrombocytopenia, and other related side effects, are correlated to an interplay of the two components in the vaccine, i.e., the spike antigen and the adenoviral vector, with the innate and immune systems, which under certain circumstances can imitate the picture of a limited COVID-19 pathological picture.

## 1. Introduction

The novel coronavirus SARS-CoV-2 (severe acute respiratory syndrome coronavirus 2), first reported in Wuhan, China, at the end of 2019 has developed into the heaviest global pandemic since the Spanish flu from 1918–1920, with more than 212 million infected persons and more than 4.4 million deaths worldwide by 23 August 2021 and 6-digit infection rates daily.

A major breakthrough in managing the COVID-19 pandemic has been the development and administration of several vaccines against SARS-CoV-2. Meanwhile, the European Medicines Agency (EMA) has approved four vaccines on the basis of randomized, blinded, and controlled trials: two messenger RNA-based vaccines, i.e., BNT162b2 (BioNTech/Pfizer) and mRNA-1273 (Moderna), that encode the spike protein antigen of SARS-CoV-2 encapsulated in lipid nanoparticles and two adenoviral vector-based vaccines, i.e., ChAdOx1 nCoV-19 (AZD1222; AstraZeneca), a recombinant chimpanzee adenoviral vector encoding the spike protein of SARS-CoV-2, and Ad26.COV2.S (Johnson & Johnson/Janssen), a recombinant human adenovirus type 26 vector encoding SARS-CoV-2 spike protein. A fifth vaccine, i.e., Gam-COVID-19-Vac (“Sputnik V”, Gamaleya National Centre of Epidemiology and Microbiology, Russia) is under evaluation by the EMA. Furthermore, an inactivated SARS-CoV-2 vaccine developed by Sinovac, China, is being applied in various countries worldwide.

The approved vaccines have been extensively tested in large clinical trials with several thousands of volunteers and show high reactivity and efficacy of protection against severe COVID-19 and generally show good safety profiles. A panel of typical side effects, including pain at the injection site, fever, chills, fatigue, and muscle pain, have been reported, but no significant numbers of severe side effects have been reported from clinical studies [1,2,3,4,5].

However, recent reports of various types of venous thrombosis, particularly of normally very rare cerebral venous sinus thrombosis (CVST), in a timely correlation to vaccination against SARS-CoV-2 with ChAdOx1 nCoV-19 have raised safety concerns. A variety of vaccine-associated thrombotic events, including cerebral venous thrombosis, splanchnic vein thrombosis, pulmonary embolism, and other thromboses, as well as disseminated intravascular coagulation, has been reported in a time frame between a few days and three weeks after ChAdOx1 nCoV-19 vaccination against SARS-CoV-2 [6,7,8].

Among the various vaccine-associated thrombotic events, special attention has been focused on normally very rare cerebral venous sinus thrombosis (CVST) in combination with pronounced thrombocytopenia, with most of these patients showing high levels of antibodies with respect to platelet factor 4–polyanion complexes without previous exposure to heparin pointing to a rare vaccine-related variant of spontaneous heparin-induced thrombocytopenia referred to as vaccine-induced immune thrombotic thrombocytopenia (VITT) [6,7,8].

A recent retrospective survey that estimated the incidence of CVST and other cerebrovascular events in temporal relation to COVID-19 vaccination with BNT162b2, ChAdOx1 nCoV-19, and mRNA-1273 in Germany showed an at least 10-fold higher CVST incidence rate in patients who received a first ChAdOx1 nCoV-19 vaccine shot compared with the highest estimate of CVST incidence rate from empirical data. Furthermore, an almost 10-fold higher risk for CVST following vaccination with ChAdOx1 nCov-19 compared to mRNA-based vaccines [9] together with recent reports on individuals who developed CVST with severe thrombocytopenia within two weeks after immunization with Ad26.COV2.S [10,11] suggests that VITT-associated thrombotic events may be associated with adenovirus vector-based vaccines coding for the SARS-CoV-2 spike protein, indicating that the underlying mechanism relies on—or at least includes—the adenoviral vectors used in this vaccine type.

The underlying mechanism of action of these thrombotic events after adenoviral vector-based SARS-CoV-2 vaccines is still unknown. The present review summarizes the published data related to thrombotic events and different hypotheses for VITT-associated thrombotic events and presents an integrated model indicating that both SARS-CoV-2 spike protein and adenovirus vector can trigger signaling pathways that individually—and at a higher probability in combination—may trigger thromboses and thrombocytopenia following vaccination.

## 2. SARS-CoV-2 and COVID-19—The Virus and the Disease

SARS-CoV-2 belongs to enveloped positive-sense, single-stranded RNA viruses, similar to the two other highly pathogenic coronaviruses such as SARS-CoV and Middle East respiratory syndrome (MERS-CoV) [12]. SARS-CoV-2 binds to the angiotensin-converting enzyme-related carboxypeptidase-2 (ACE-2) receptor on target cells by its spike (S) protein. The spike (S) protein is composed of the S1 subunit containing the highly conserved receptor binding domain (RBD) and the S2 subunit, which mediates fusion between the viral and host cell membranes after cleavage by the cellular serine protease TMPRSS2 [13]. This furin-like cleavage site is unique to the S protein of SARS-CoV-2 and may, together with the particularly high binding affinity to the target receptor and the peculiarity of a long symptom-free but nevertheless highly infectious time period between infection and appearance of first symptoms or asymptomatic transmission, be responsible for the particularly efficient spread of SARS-CoV-2 compared to previous pathogenic hCoVs [14]. The ACE-2 receptor is widely expressed in pulmonary and cardiovascular tissues and hematopoietic cells, including monocytes and macrophages, which may explain the broad range of pulmonary and extrapulmonary effects of SARS-CoV-2 infection, including cardiac, gastrointestinal organs, and kidney affection [13,15,16].

The majority of individuals infected with SARS-CoV-2 show mild-to moderate symptoms, and up to 20% of infections may be asymptomatic. Symptomatic patients show a wide spectrum of clinical manifestations ranging from mild febrile illness and coughing up to acute respiratory distress syndrome (ARDS), multiple organ failure, and death. Thus, the clinical picture of severe cases is very similar to that observed in SARS-CoV- and MERS-CoV-infected patients [12]. While younger individuals show predominantly mild-to-moderate clinical symptoms, elderly individuals frequently exhibit severe clinical manifestations [17,18,19,20,21,22,23]. Pre-existing comorbidities, including diabetes, respiratory and cardiovascular diseases, renal failure and sepsis, older age, and male sex, seem to be associated with more severe disease and higher mortality [20,21,22,23,24]. Postmortem analysis of fatal COVID-19 showed diffuse alveolar disease with capillary congestion, cell necrosis, interstitial edema, platelet-fibrin thrombi, and infiltrates of macrophages and lymphocytes [25]. Furthermore, the induction of endotheliitis in various organs (including lungs, heart, kidney, and intestine) by SARS-CoV-2 infection as a direct consequence of viral involvement and of the host inflammatory response has been demonstrated [15,16].

The molecular mechanisms for the morbidity and mortality of SARS-CoV-2 are still incompletely understood. Virus-induced cytopathic effects and viral evasion of the host immune response, in particular, the inhibition of the host IFN type I response by SARS-CoV-2 [26], seem to play a role in disease severity. Furthermore, clinical data from patients, particularly those with severe clinical manifestations, indicate that highly dysregulated exuberant inflammatory and immune responses correlate with the severity and lethality of disease [15,16,18,27,28,29]. Significantly elevated cytokine and chemokine levels, also termed “cytokine storm”, are assumed to play a central role in the severity and lethality of SARS-CoV-2 infections. Upregulated plasma levels of IL-1β, IL-7, IL-8, IL-9, IL-10, G-CSF, GM-CSF, IFNγ, IP-10, MCP-1, MIP-1α, MIP-1β, PDGF, TNFα, and VEGF have been reported in both ICU (intensive care unit) patients and non-ICU patients. Notably, significantly higher plasma levels of IL-2, IL-7, IL-10, G-CSF, IP-10, MCP-1, MIP-1α, and TNFα were found in patients with severe pneumonia developing ARDS and required ICU admission and oxygen therapy compared to non-ICU patients showing pneumonia without ADRS [18]. Various studies have shown that highly stimulated epithelial-immune cell interactions result in exuberant dysregulated inflammatory responses with significantly (topically and systemically) elevated cytokine and chemokine release [30,31].

Regarding the underlying signaling pathways, recent data suggest that the NF-κB pathway is one of the central signaling pathways for the SARS-CoV-2 infection-induced proinflammatory cytokine/chemokine response, playing a central role in the severity and lethality of COVID-19 [32,33,34,35,36,37,38]. This NF-κB-triggered proinflammatory response in acute COVID-19 is shared with other acute respiratory viral infections caused by a highly pathogenic influenza A virus of H1N1 (e.g., Spanish flu) and H5N1 (avian flu) origin, SARS-CoV, and MERS-CoV [38].

Excessive activations of exuberant inflammatory responses with involvement of endothelial cells, epithelial cells, and immune cells are assumed to result in further disturbances in a variety of other integrated systems, including the complement system, coagulation, and bradykinin systems, resulting in increased coagulopathies and feeding back into positive signaling feedback loops accelerating COVID-19-associated inflammatory processes [39,40,41,42,43,44]. In particular, vascular occlusions by neutrophil extracellular traps (NETs) and disturbances of coagulation with various types of thromboses, multiple microthromboses, thrombocytopathy, and endotheliopathy seem to be another hallmark of the COVID-19 disease, and the development of coagulopathies is one of the key and persistent features associated with poor outcome. In particular, elevated D-dimer levels, prolonged prothrombin time, thrombocytopenia, and low fibrinogen (indicating fibrinogen consumption) have been found to be prognostic indicators for poor outcome [45,46,47,48,49,50]. Lung histopathology often reveals fibrin-based blockages in the small blood vessels of patients who succumb to COVID-19 [25].

Furthermore, various types of antiphospholipid (aPL) antibodies targeting phospholipids and phospholipid-binding proteins, including anti-cardiolipin IgG, IgM, and IgA; anti-β2 glycoprotein I IgG, IgM, and IgA; and anti-phosphatidylserine/prothrombin (aPS/PT) IgG, and IgM, were found in 52% of serum samples from 172 patients hospitalized with COVID-19. Higher titers of aPL antibodies were associated with neutrophil hyperactivity, including the release of neutrophil extracellular traps (NETs), higher platelet counts, more severe respiratory disease, and lower clinical estimated glomerular filtration rate. Similar to IgG from patients with anti-phospholipid syndrome, IgG fractions isolated from patients with COVID-19 promoted NET release from neutrophils isolated from healthy individuals. Furthermore, the injection of IgG purified from COVID-19 patient serum into mice accelerated venous thrombosis as shown in two mice models. These findings suggest that half of the patients hospitalized with COVID-19 become at least transiently positive for aPL antibodies and that these autoantibodies are potentially pathogenic [51]. High rates of thrombosis and thrombotic-related complications have been reported in adult patients with severe COVID-19 as well as in children developing COVID-19 or multisystem inflammatory syndrome (MIS-C). Studies in adults have invoked thrombotic microangiopathy (TMA) as a potential cause for severe manifestations of COVID-19 [52,53,54]. TMA results from endothelial cell damage to small blood vessels, resulting in hemolytic anemia, thrombocytopenia, and organ damage in some cases [55,56,57,58,59]. TMA has been reported in postmortem studies of adult patients with COVID-19 [60]. Regarding therapeutic intervention, a retrospective analysis examined the association of in-hospital anticoagulation procedures with mortality, intubation, and major bleeding. In-hospital anticoagulation was associated with lower mortality and intubation among hospitalized COVID-19 patients [61].

## 3. Frequently Described Side Effects after Vaccination against SARS-CoV-2

Before certification by the regulatory authorities, i.e., FDA and EMA, respectively, vaccines have been tested in thousands of volunteers in large clinical trials [1,2,3,4,5,62]. A panel of typical side effects has been reported, such as short-term, mild-to-moderate pain, redness, and swelling at the injection site and systemic flu-like symptoms, including fatigue, headache, muscle pain, chills, joint pain, and fever, and were shown to appear in varying degrees for all vaccines. There were no major differences between the mRNA vaccine and the adenoviral vector vaccines, with one exception, i.e., that the mRNA vaccine (i.e., BNT162b2 and mRNA-1273) showed more pronounced side effects after the second immunization, whereas the adenoviral vector vaccines showed more pronounced side effects after the first immunization, with a lower intensity/prevalence of side effects after the second vaccination for the ChAdOx1 nCoV-19 vaccine (AZD1222). Furthermore, younger individuals (<55 years) generally showed a higher incidence and intensity of side effects compared to aged persons (>55 years) reported for both types of vaccines, mRNA and adenoviral vector-based.

Importantly, no significant increase in the prevalence of thrombotic events has been reported during clinical studies, with large phase 3 trials tested in more than 30,000–40,000 volunteers for various vaccines [1,2,3,4,5,62].

## 4. Rare Serious Side Effects after Vaccination Visible during Mass Vaccination after Market Approval

Following mass vaccination after market approval, recent reports of cerebral venous sinus thrombosis (CVST) and a variety of other thrombotic events after ChAdOx1 vaccination against SARS-CoV-2 have raised safety concerns.

One study recently published in *N. Engl. J. Med.* showed venous thrombosis and thrombocytopenia seven to 10 days after receiving the first dose of the ChAdOx1 nCoV-19 adenoviral vector vaccine against coronavirus disease 2019 (COVID-19). All patients had high levels of antibodies relative to platelet factor 4 (PF4)–polyanion complexes without previous exposure to heparin, pointing to a rare vaccine-related variant of spontaneous heparin-induced thrombocytopenia referred to as vaccine-induced immune thrombotic thrombocytopenia [6]. A second study published in *N. Engl. J. Med.* assessed the clinical and laboratory characteristics of patients who had developed thrombosis or thrombocytopenia after vaccination with ChAdOx1 nCov-19 or other vaccine-associated thrombotic events, including nine cerebral venous thrombosis, three splanchnic vein thrombosis, three pulmonary embolism, four other thromboses, and five disseminated intravascular coagulation. All 28 patients tested positive for antibodies against PF4–heparin and tested positive in a platelet-activation assay in the presence of PF4 independent of heparin. Platelet activation was inhibited by high levels of heparin, Fc receptor-blocking monoclonal antibody, and immune globulin. Additional studies with PF4 or PF4–heparin affinity purified antibodies in two patients confirmed PF4-dependent platelet activation. The authors concluded that vaccination with ChAdOx1 nCov-19 can result in the rare development of immune thrombotic thrombocytopenia mediated by platelet-activating antibodies against PF4, which clinically mimics autoimmune heparin-induced thrombocytopenia [7]. A third study published in *N. Engl. J. Med.* showed similar results in 22 patients who presented with thrombocytopenia and thrombosis, primarily cerebral venous thrombosis, and one patient with isolated thrombocytopenia and a hemorrhagic phenotype. All of the patients had low or normal fibrinogen levels and strongly increased D-dimer levels including 22 patients who were positive for antibodies against PF-4, and one patient was negative [8]. Meanwhile, in several European countries, cases of venous thrombosis, including cerebral venous sinus thrombosis (CVST), have been reported in the temporal context with ChAdOx1 nCov-19 vaccine administration. At the beginning of March 2021, 30 venous thromboembolic events were reported to EMA out of approximately 5 million persons who had received the ChAdOx1 nCoV-19 vaccine at that time [9]. The UK’s Medicines and Healthcare Products Regulatory Agency had received 79 reports of thrombosis associated with low platelets by 31 March, of which 44 were CVST. Of these 79 cases, 51 (13 fatal) were women, and 28 (six fatal) were men. All of the UK cases occurred after the first dose. The risk was higher in the younger age groups, starting at 1.1 serious harm events for 100,000 immunized people among those aged 20–29 years and falling to 0.2/100,000 in those aged 60–69. These events were recorded within a time interval of less than 4 weeks after vaccination. For comparison, in women taking hormonal contraceptives, the risk of thrombosis is approximately 60/100,000 person years, and the risk of fatal pulmonary embolism is approximately 1/100,000. Furthermore, several cases of serious thrombosis with thrombocytopenia have been reported after the use of the Johnson & Johnson (Janssen) COVID-19 vaccine [63].

A recent retrospective survey estimated the incidence of CVST and other cerebrovascular events in temporal relation to COVID-19 vaccination with BNT162b2, ChAdOx1 nCov-19, and mRNA-1273 in Germany. According to this study up to 14 April 2021, Germany identified 62 vascular cerebrovascular adverse events in close temporal relationship with a COVID-19 vaccination, of which 45 cases were CVST. Eleven patients died. The authors estimated an incidence rate of CVST within one month from first dose administration of 17.9 per 100,000 person-years for ChAdOx1 nCov-19 vaccine and 1.3 per 100,000 person-for BNT162b2. Before the COVID-19 pandemic, the incidence rate of CVST was estimated to be between 0.22 and 1.75 per 100,000 person-years in four European countries, Australia, Iran, and Hong Kong. Accordingly, a 10-fold to 90-fold higher CVST incidence rate in patients who received a first dose of the ChAdOx1 nCov-19 vaccine was observed compared with the highest or lowest estimate of the CVT incidence rate from empirical data, respectively. The incidence rate of a CVST event after the first dose COVID-19 vaccination was also statistically significantly increased for ChAdOx1 nCov-19 compared to mRNA-based vaccines (9.68 and 3.46 to 34.98) and for females compared to non-females (3.14 and 1.22 to 10.65) [9]. The 10-fold higher risk for CVST following vaccination with ChAdOx1 nCov-19 compared to mRNA-based vaccines together with recent reports on individuals who developed CVST with severe thrombocytopenia within two weeks after immunization with Ad26.COV2.S [10,11,63] suggests that VITT-associated thrombotic events may be associated with adenovirus vector-based vaccines coding for the SARS-CoV-2 spike protein, indicating that the mechanism of action relies or at least includes the adenoviral vector used in this vaccine. The underlying mechanism of action of these thrombotic events after adenoviral vector-based SARS-CoC-2 vaccines is still not completely known, although various data and hypotheses have been raised (see below).

In addition to vaccine-associated thrombotic events, several additional serious conditions have been reported in association with vaccination against SARS-CoV-2, including single cases of capillary leakage syndrome and coronary myocarditis after immunization with ChAdOx1 nCoV-19 (AZD1222) (AstraZeneca, Cambridge, UK) and BNT162b2 (BioNTech/Pfizer, Mainz, Germany/New York, USA), respectively.

## 5. Superantigen Hypothesis

To elucidate the underlying mechanisms of various rare side effects following anti-SARS-CoV-2 vaccination, one particular disease pattern appearing in a timely context with respect to the SARS-CoV-2 pandemic may be of interest. During the COVID-19 pandemic, a new deadly disease in children named multisystem inflammatory syndrome in children (MIS-C) has received much attention, which rapidly progresses to hyperinflammation and shock and can result in multiple organ failure in a high percentage of affected children. MIS-C has been found to be temporally associated with the COVID-19 pandemic with a few weeks delay following peaks in SARS-CoV-2 infection incidence and was often associated with SARS-CoV-2 exposure or the presence of SARS-CoV-2 reactive antibodies in affected children. After initial reports in the UK, an increasing number of cases have been reported in Europe and New York a few weeks after epidemic peaks. MIS-C manifests as persistent high fever and hyperinflammation with multiorgan system involvement, including cardiac, gastrointestinal, renal, hematologic, dermatologic, and neurologic symptoms. The overall clinical picture of MIS-C, however, is often similar in many aspects to the late, severe COVID-19 phase in adults, characterized by cytokine storm, hyperinflammation, multiorgan damage, severe myocarditis, and acute kidney injury [64]. A causal link between SARS-CoV-2 infection and MIS-C has not yet been firmly established; however, many patients with MIS-C were reportedly exposed to someone known or suspected to have COVID-19. Furthermore, only approximately one-third of patients with MIS-C tested positive for SARS-CoV-2 by PCR, but a large majority of PCR-negative patients were serologically positive for SARS-CoV-2 antibodies and/or had a history of mild COVID-19 infection or exposure several weeks before presentation. This timing suggests that MIS-C is a postinfectious disease or an immune or autoimmune disease triggered by SARS-CoV-2 infection. Although MIS-C was initially observed to resemble Kawasaki disease (KD), clinical and laboratory characteristics indicate that MIS-C is rather reminiscent of toxic shock syndrome (TSS), which is usually found in severe cases after sepsis with Gram-positive bacteria such as *Staphylococcus aureus* or *Streptococcus pyogenes,* as indicated by typical gastrointestinal involvement, myocardial dysfunction and cardiovascular shock, pronounced lymphopenia and thrombocytopenia, and high coagulation parameters, such as D-dimers, found in MIS-C and TSS but typically not in KD [64]. Notably, TSS is known to be caused by different types of superantigens (SAgs), including bacteria and viruses, with bacterial SAgs being broadly studied. They include proteins secreted by *Staphylococcus aureus* and *Streptococcus pyogenes* that stimulate massive production of inflammatory cytokines and toxic shock. Typical examples are toxic shock syndrome (TSS) toxin 1 and staphylococcal enterotoxins B (SEB) and H (SEH). They are highly potent T cell activators that can bind to major histocompatibility complex (MHC) class II (MHCII) molecules and/or directly to T cell receptors (TCRs) of both CD4+ and CD8+ T cells. The ability of SAgs to bypass the antigen specificity of TCRs results in broad activation of T cells and a cytokine storm, resulting in toxic shock. Notably, SAgs do not bind the major (antigenic) peptide-binding groove of MHCII but instead bind other regions or αβ TCRs directly [64].

Importantly, by using structure-based computational models, Cheng et al. demonstrated that the SARS-CoV-2 spike (S) glycoprotein exhibits a high-affinity motif for binding TCRs and may form a ternary complex with MHCII. The binding epitope on the spike protein harbors a sequence motif unique to SARS-CoV-2 (which is not present in other SARS-related coronaviruses), which is highly similar in both sequence and structure to the bacterial superantigen staphylococcal enterotoxin B. Furthermore, the interfacial region includes selected residues from an intercellular adhesion molecule (ICAM)-like motif shared between the SARS viruses from the 2003 and 2019 pandemics. A neurotoxin-like sequence motif on the receptor-binding domain also exhibits a high tendency to bind TCRs. An analysis of the TCR repertoire in adult COVID-19 patients demonstrated that those with severe hyperinflammatory disease exhibit TCR skewing consistent with superantigen activation [65]. A blood test that determines the presence of specific TCR variable gene segments for the identification of patients at risk for severe MIS-C has been developed [66]. These data suggest that the SARS-CoV-2 spike protein itself may act as a superantigen to trigger the development of MIS-C as well as cytokine storm in adult COVID-19 patients [67]. Interestingly, the first cases of MIS following SARS-CoV-2 infection in adults have been reported [68].

### 5.1. Superantigen Induce Procoagulant Activity

In the context of the superantigen hypothesis, it is noteworthy that, during severe sepsis, activation of blood coagulation plays a critical pathophysiological role resulting in septic shock, microthrombi, and multiorgan dysfunction. During severe sepsis and septic shock, a massive release of cytokines and activation of the coagulation system result in disseminated intravascular coagulation (DIC) and multiorgan dysfunction syndrome [69,70,71]. The procoagulant activities and tissue factor induction by various superantigens from *Staphylococcus aureus*, including enterotoxin A (EA), enterotoxin B (EB), and toxic shock syndrome toxin (TSST)-1, were tested for their ability to induce procoagulant activity and tissue factor (TF) expression—a major initiator of blood coagulation expressed predominantly on monocytic cells and endothelial cells—in human whole blood and in peripheral blood mononuclear cells. The determination of clotting time showed that enterotoxin A, B, and toxic shock syndrome toxin 1 from *Staphylococcus aureus* induced procoagulant activity in whole blood and in mononuclear cells. Procoagulant activity was dependent on the expression of TF in monocytes. In the supernatants from staphylococcal toxin-stimulated mononuclear cells, interleukin (IL)-1 beta was detected by ELISA. The increased procoagulant activity and TF expression in monocytes induced by staphylococcal toxins were inhibited in the presence of IL-1 receptor antagonist, which is a natural inhibitor of IL-1 beta. The study demonstrated that superantigens from *Staphylococcus aureus* activate the extrinsic coagulation pathway by inducing the expression of TF in monocytes and that the expression is mainly triggered by superantigen-induced IL-1 beta release [70,71].

### 5.2. Superantigens and NF-kappaB

Furthermore, toxic shock syndrome toxin-1 (TSST-1) and staphylococcal enterotoxins A and B induce the activation of NF-κB, which acts as a transcriptional enhancer by binding to sequences found in both the IL-1 beta and TNF-alpha promoters. The induction of both NF-κB DNA-binding proteins and NF-κB enhancer function was downregulated by inhibitors of protein kinase C and protein tyrosine kinase, indicating a role for these protein kinases in the induction of NF-κB by MHC class II ligands. By using neutralizing antibodies, it was demonstrated that after the stimulation of cells with TSST-1, TNFα acted to upregulate the binding of NF-κB to DNA and the activation of the NF-κB promoter CAT construct, indicating that the induction of NF-κB by superantigens is upregulated in part by an autocrine loop involving TNFα [72].

The central role of the NF-κB pathway in superantigen-mediated T cell activation has also been demonstrated in studies showing that proteasome inhibition reduced superantigen-mediated T cell activation. PS-519, a potent and selective proteasome inhibitor, was shown to inhibit NF-κB activation by blocking the degradation of its inhibitory protein IκB and reducing superantigen-mediated T cell activation in vitro and in vivo. Proliferation was inhibited along with the expression of very early (CD69), early (CD25), and late T cell (HLA-DR) activation molecules. Moreover, the expression of E-selectin ligands relevant to dermal T cell homing was reduced, as was E-selectin binding in vitro [73]. Furthermore, the inhibition of the NF-κB pathway by two antioxidants, N-acetyl-cysteine (NAC) and pyrrolidine dithiocarbamate (PDTC), was shown to dose-dependently inhibit SE-stimulated T-cell proliferation (by 98%), the production of cytokines and chemokines by PBMCs, and the expression of SE-induced cell surface activation markers. The potency of both NAC and PDTC corresponded to their ability to inhibit NF-κB activation [74]. In addition to the MHCII-dependent activation of T-cells, MHC/II-independent direct stimulation of TCR Vb by Staphylococcus aureus enterotoxin via PKCtheta (PKCθ)/NF-κB and IL2R/STAT signaling pathways has also been shown [75].

## 6. NF-κB Pathway

### 6.1. NF-κB Pathway Is Part of Normal T Cell Activation

In this context, it must be mentioned that normal or physiological antigen stimulation of TCR signaling to NF-κB is required for T cell proliferation and differentiation of effector cells. The engagement of the TCR by an MHC-antigen complex initiates an entire chain of downstream events, described in detail by Paul et al., which ultimately trigger calcium release and PKC activation, respectively. Activation of a specific PKC isoform, PKCθ, connects TCR proximal signaling events to distal events that ultimately result in NF-κB activation. Importantly, PKCθ activation is also driven by engagement of the T cell costimulatory receptor CD28 by B7 ligands on antigen presenting cells and via intermediate steps resulting in the activation of IKKβ. IKKβ then phosphorylates IκBα, triggering its proteasomal degradation and enabling nuclear translocation of canonical NF-κB heterodimers composed of p65 (RELA) and p50 proteins. Once in the nucleus, NF-κB governs the transcription of numerous genes involved in T cell survival, proliferation, and effector functions [76].

### 6.2. SARS-CoV-2 Spike Protein Induces NF-κB

Importantly, the pathogenesis of COVID-19 has been shown to involve overactivation of the NF-κB pathway [37,38]. In several studies, the question of which part(s) of SARS-CoV-2 is responsible for massive NF-κB pathway activation has been studied. Khan et al. investigated the direct inflammatory functions of major structural proteins of SARS-CoV-2 and showed that the spike (S) protein potently induces inflammatory cytokines and chemokines, including IL-6, IL-1β, TNFα, CXCL1, CXCL2, and CCL2 but not IFNs in human and mouse macrophages. No such inflammatory response was observed in response to membrane (M), envelope (E), or nucleocapsid (N) proteins. When stimulated with extracellular S protein, A549 human lung epithelial cells also produced inflammatory cytokines and chemokines. Interestingly, epithelial cells expressing S protein intracellularly are non-inflammatory but elicit an inflammatory response in macrophages when cocultured. Biochemical studies revealed that the S protein triggers inflammation via activation of the NF-κB pathway in a MyD88-dependent manner. Furthermore, the activation of the NF-κB pathway was abrogated in TLR2-deficient macrophages. Consistently, administration of the S protein induced IL-6, TNFα, and IL-1β in wild-type but not TLR2-deficient mice. In this study, both S1 and S2 were demonstrated to show high NF-κB activation, with S2 showing higher potency on an equimolar basis [77]. In a second study, the spike protein was demonstrated to promote an angiotensin II type 1 receptor (AT1)-mediated signaling cascade, induced transcriptional regulatory molecules NF-κB and AP-1/c-Fos via MAPK activation, and increased IL-6 release [78]. A third study demonstrated that SARS-CoV-2 spike protein subunit 1 (CoV2-S1) induces high levels of NF-κB activation, production of proinflammatory cytokines, and mild epithelial damage in human bronchial epithelial cells. CoV2-S1-induced NF-κB activation requires S1 interaction with the human ACE2 receptor and early activation of endoplasmic reticulum (ER) stress and associated unfolded protein response (UPR) and MAP kinase signaling pathways. The FDA-approved ER stress inhibitor 4-phenylburic acid (4-PBA) and MAP kinase inhibitors trametinib and ulixertinib ameliorated CoV2-S1-induced inflammation and epithelial damage [79]. Furthermore, the effect of recombinant SARS-CoV-2 spike protein S1 on human peripheral blood mononuclear cells (PBMCs) was studied. PBMCs were stimulated with spike S1 protein resulting in a significant release of TNFα, IL-6, IL-1β, and IL-8. This cytokine release was inhibited by pretreatment with dexamethasone. Moreover, S1 stimulation of PBMCs activated the NF-κB pathway as demonstrated by phosphorylation of NF-κB p65, IκBα degradation, and increased DNA binding of NF-κB p65 after stimulation with spike S1 protein. NF-κB activation was blocked by treatment of PBMCs with dexamethasone or the specific NF-κB inhibitor BAY11-7082. The activation of p38 MAPK by spike S1 protein was blocked by dexamethasone or SKF 86002. Release of TNFα, IL-6, IL-1β, and IL-8 was reduced in the presence of BAY11-7082 or SKF 86002, while pretreatment with the NLRP3 inhibitor CRID3 reduced IL-1β production. The data indicate that SARS-CoV-2 spike protein S1 stimulates PBMCs to release proinflammatory cytokines via mechanisms involving the activation of NF-κB, p38 MAPK, and NLRP3 inflammasomes [80].

Furthermore, the interaction between the SARS-CoV-2 spike (S) protein and LPS was investigated. SARS-CoV-2 S protein was shown to bind to LPS. Spike protein, when combined with low levels of LPS, boosted NF-κB activation in monocytic THP-1 cells and cytokine responses in human blood and PBMC, respectively. The study demonstrated that the S protein modulated the aggregation state of LPS, providing a potential molecular link between excessive inflammation during infection with SARS-CoV-2 and comorbidities involving increased levels of bacterial endotoxins [81].

Purified recombinant S protein was studied to stimulate murine macrophages (RAW264.7) to produce proinflammatory cytokines (IL-6 and TNFα) and the chemokine IL-8. The authors found direct inductions of IL-6 and TNF release in the supernatant in a dose-dependent and time-dependent manner, and they were highly spike protein specific. IL-6 and TNF productions are dependent on NF-κB, which is activated by IκB degradation [82].

In a mouse model, the cleaved S1 subunit of the spike protein was demonstrated to elicit strong pulmonary and systemic inflammatory responses in transgenic K18-hACE2 mice and milder responses in wild type mice following intratracheal installation, which was accompanied by loss in body weight, dramatically increased white blood cell count and protein concentrations in bronchoalveolar lavage fluid (BALF), and upregulation of multiple inflammatory cytokines, including INFγ, IL-1β, IL-6, IL-17, monocyte chemoattractant protein-1 (MCP-1), keratinocytes-derived chemokine (KC, CXCL1), TNFα, macrophage inflammatory proteins MIP1α and MIP1β, and IFNγ-inducible protein 10 (IP-10) in BALF and serum. Furthermore, the activations of NF-κB and of the signal transducer and activator of transcription 3 (STAT3) were demonstrated [83].

Overall, these data show that the SARS-CoV-2 S spike protein induces powerful NF-κB activation, showing strong similarity to data recorded for the SARS-CoV S protein. Similar to SARS-CoV-2, the clinical picture of severe acute respiratory syndrome (SARS) is characterized by an overexuberant immune response with lung lymphomononuclear cell infiltration and proliferation that may account for tissue damage more than the direct effect of viral replication.

### 6.3. NF-κB Activation Is Central to Coagulation Events

The effects of NF-κB pathway activation on various coagulopathies may be relevant in VITT.

Plasminogen activator inhibitor-1 (PAI-1) is the major inhibitor of plasminogen activation and likely plays important roles in coronary thrombosis and arteriosclerosis. Tumor necrosis factor-alpha (TNFα) is one of many recognized physiological regulators of PAI-1 expression and may contribute to elevated plasma PAI-1 levels in sepsis and obesity. A 5′ distal TNFα-responsive enhancer of the PAI-1 gene is located 15 kb upstream of the transcription start site containing a conserved NF-κB-binding site that mediates the response to TNFα. This newly recognized site was fully capable of binding NF-κB subunits p50 and p65, whereas the overexpression of the NF-κB inhibitor IkappaB prevents TNFα-induced activation of this enhancer element [84].

In another study, the development of procoagulant activity and monocyte activation in heparinized whole blood during extracorporeal circulation was studied. Anaphylatoxins C3a and C5a appeared in the blood within 30 min of circulation. Circulated blood developed a marked potential for coagulation that reached maximal activity by 4 h of circulation. This procoagulant activity was neutralized by anti-tissue factor antibody. Isolation of monocytes from circulated blood showed increased tissue factor (TF) expression on the cell surface. Furthermore, NF-κB nuclear translocation was found in monocytes from blood passing through the circuit, suggesting that TF expression was due to monocyte stimulation and transcriptional activation of the TF gene. The upregulated TF expression was followed by a 30-fold increase in thrombin generation. Monocyte NF-κB activation, monocyte TF expression, thrombin generation, and the procoagulant activity of blood in extracorporeal circulation were all blocked by the proteasome inhibitor MG132, indicating that intravascular TF expression during extracorporeal circulation of blood is due to NF-κB-mediated activation of monocytes (possibly by complement) [85].

Complement components C3a and C5a were studied regarding their effect on plasminogen activator inhibitors (PAI-1) in human macrophages. C5a increased PAI-1 (both mRNA and protein level) in human monocyte-derived macrophages. Pertussis toxin or anti-C5aR/CD88 antibody completely abolished the effect of recombinant human C5a on PAI-1 induction, indicating a role of the C5a receptor. Furthermore, C5a induced NF-κB binding and the increase in PAI-1 were completely abolished by an NF-κB inhibitor, indicating that C5a upregulates PAI-1 in macrophages via NF-κB activation [86].

Finally, platelets are megakaryocyte-derived fragments lacking nuclei and prepped to maintain primary hemostasis by initiating blood clots on injured vascular endothelia. Pathologically, platelets undergo the same physiological processes of activation, secretion, and aggregation with such pronouncedness that they orchestrate and make headway in terms of the progression of atherothrombotic diseases not only through clot formation but also by forcing a pro-inflammatory state. Indeed, NF-κB has been implicated in platelet survival and function [87].

## 7. SARS-CoV-2 Spike Protein Subunit S1 Induces JAK/STAT3 Activation

Parallel to the activation of the NF-κB pathway (or probably also as a result of the NF-κB-induced cytokines/chemokines), there is also a prominent activation of Janus kinases/signal transducer and activator of transcription 3 (JAK/STAT3) pathway following SARS-CoV-2 infection. Cytokines which bind to type I (e.g., IL-6, G-CSF) and type II (e.g., IL-10) receptors, as well as multiple chemokines binding to G-protein coupled receptors, directly trigger the activation of JAK/STAT3 pathway, characterized by Tyr phosphorylation [88,89].

Importantly, STAT3 activation was shown to be involved in COVID-19 associated coagulopathy via enhancement of the expression of tissue factor and of PAI-1 [90].

Activation of STAT3 and IL-6, which are potent activators of the JAK/STAT3 pathway, have been described to be involved in coagulation and complement activation [50,89,90,91].

Various other cytokines, such as G-SCF or GM-CSF, and a variety of chemokines such as IL-8, MCP-1, MIP1α, and MIP1β found to be elevated in acute COVID-19 patients [18] are also known to activate the JAK/SAT3 pathway [88,92]. The enhanced transcription of JAK/STAT3 activators, such as IL-6 and multiple chemokines (including IL-8, MCP-1, MIP1α, and MIP1β), found during acute respiratory viral infections, including also SARS-CoV-2, is largely triggered also by NF-κB pathway activation [38,93,94]. Importantly, a positive feedback loop of IL-6–JAK/STAT3 signaling with amplification of NF-κB activation was suggested to be involved in COVID-19 induced cytokine storm and mortality [33,34].

By using the transgenic K18-hACE2 mouse model, it was shown that not only SARS-CoV-2 infection but also the cleaved S1 subunit of the spike protein elicited strong pulmonary and systemic inflammatory responses after intratracheal application, accompanied by dramatically increased white blood cell count and protein concentration in BALF and upregulation of multiple inflammatory cytokines in BALF and serum. Notably, the activation of NF-κB pathway and phosphorylation of the signal transducer and activator of transcription 3 (STAT3) were found [83].

## 8. Platelet Factor 4

Vaccine-induced thrombotic thrombocytopenia (VITT)*,* as observed in rare cases following vaccination with vaccines from AstraZeneca or Johnson & Johnson, resembled heparin-induced thrombocytopenia (HIT). The hallmarks of HIT are antibodies specific for the heparin/platelet factor 4 (PF4) complex that cause thrombocytopenia and thrombosis through platelet activation.

Platelet factor 4 (PF4 and CXCL4) released by activated platelets belongs to the chemokine family and has essential functions in blood coagulation. PF4 represents a positively charged tetramer with a high binding affinity for heparin and other glycosaminoglycans (GAGs). Binding of the positively charged PF4 results in the neutralization of the negative charge of the heparan sulfate side chains of GAGs on the surface of platelets and endothelial cells, which facilitates platelet aggregation and formation of thrombus. In addition to blood coagulation, PF4 has additional activities, with elevated PF4 expression found after trauma and in response to infection [95].

Importantly, when patients are exposed to heparin, heparin binds to PF4 and promotes PF4 aggregation, forming ultra large and antigenic PF4–heparin complexes. A certain percentage of patients develop antibodies against PF4–heparin complexes that cause heparin-induced thrombocytopenia (HIT). Anti-PF4–heparin antibodies are found in approximately half of patients after cardiac surgery and in more than 10% of surgical patients treated with heparin compared to 3–4% of healthy subjects. However, only 5–30% of patients with antibodies against heparin develop HIT [95]. Circulating PF4–heparin-antibody complexes can bind to the FcγRIIA receptor on platelets and to a variety of additional Fc receptor-bearing blood cells, such as monocytes and neutrophils. Binding to FcγRIIA causes platelet activation, resulting in the release of the contents of the cytoplasmic granules and procoagulant microparticles. The activation of platelets and neutrophils by HIT antibodies can activate the vascular endothelium. PF4–heparin immune complexes can also directly activate endothelial cells by enhancing the expression of adhesion molecules such as P-selectins and E-selectins and the release of von Willebrand factor. Platelet activation after binding of HIT-related immune complexes to FcγRIIA and transactivation of monocytes and endothelial cell increase also the expression of phosphatidylserine (PS) and the binding of factor Xa to platelets, which results in the generation of thrombin and thrombotic vessel occlusions, e.g., venous thromboembolism [96]. Binding of antibodies to PF4–heparin complexes occur only at approximately equimolar ratios, i.e., 1 mol PF4 tetramer to 1 mol unfractionated heparin (UFH). At equimolar ratios, PF4 and UFH form ultra large complexes that are highly antigenic and promote antibody binding and platelet activation. In contrast, low molecular weight heparin (LMWH) is less antigenic and forms ultra large complexes less efficiently. The bindings of HIT antibodies to platelets and the induction of thrombocytopenia are proportional to PF4 expression. Heparin, in particular UFH, increases the severity of thrombocytopenia. In contrast, very high doses of heparin disturb the optimal ratio and disrupt antigen formation, preventing thrombocytopenia induced by HIT antibody [97].

A most striking observation, however, was that heparin-naive patients can also generate anti-PF4–heparin IgG antibodies starting from day 4 following heparin treatment, which suggests pre-immunization by antigens that mimic PF4–heparin complexes. This pre-immunization was suggested to be caused by bacterial infections, supported by data on charge-dependent binding of PF4 to various bacteria, cross reaction of human heparin-induced anti-PF4–heparin antibodies with PF4-coated *Staphylococcus aureus* and *Escherichia coli*, and the appearance of anti-PF4/heparin antibodies during bacterial sepsis without heparin application in mice. Induced antibodies can bind to a large variety of PF4-coated bacteria and enhance bacterial phagocytosis in vitro. Similar antigenic epitopes are expressed when heparin binds to platelets, augmenting the formation of PF4 complexes. Boosting of preformed B cells by PF4–heparin complexes could explain the early occurrence of IgG antibodies in HIT. A continuous distribution of anti-PF4–heparin IgM and IgG serum concentrations in a representative epidemiological study was found, indicating frequent preimmunization events directed against modified PF4. Together, these data indicate that PF4 may have an important role in bacterial defense, and HIT may only be a misdirected antibacterial host defense mechanism [98].

We hypothesized that cationic PF4 may charge-dependently associate not only with bacterial surface structures but also with viruses harboring a negative surface charge, such as adenoviruses. Due to the highly negatively charged hexone protein—which represents the major capsid protein—adenovirus exposes a highly negative surface charge [99,100]. In this context, complexation with polycations has been used to enhance the transfection efficacy of adenoviral vectors into target cells. It has been demonstrated that a cationic component can charge-associate with adenovirus particles, which carry a net negative surface charge and will facilitate attachment to the negatively charged cell membrane—an approach employed for increasing the gene transfer efficacy of adenoviral vectors [101]. Similarly, adsorption of adenovirus coding for the *beta-gal2* gene (Ad2 beta gal2) in the presence of different polycations, such as polybrene, protamine, DEAE-dextran, or poly-L-lysine, significantly increased transfection efficacy into various cell types. The poly-anion heparin completely abrogated the effects of polycations [102]. Finally, a recent bioRxiv preprint revealed the structure of ChAdOx1/AZD-1222. ChAdOx1 shares the archetypal icosahedral, T = 25, capsid common to adenoviruses. Binding of fiber knobs to the coxsackie and adenovirus receptor (CAR) was shown to be the primary mechanism ChAdOx1 uses to attach to cells. Furthermore, the work revealed that the surface of the ChAdOx1 viral capsid has strong electronegative potential. The ChAdOx1 hexon hypervariable regions (HVRs) are different from other adenoviruses. In terms of apical electrostatic surface potential, calculated on equilibrated hexon structures, ChAdOx1 has the most electronegative surface potential, which can be expected to influence the strength of incidental charge-based interactions with other molecules. Molecular simulations have suggested that this charge, together with shape complementarity, is a mechanism by which an oppositely charged protein, e.g., platelet factor 4 (PF4), may bind the vector surface [103].

While a previous study by Greinacher’s team has shown no cross-reactivity of platelet-activating antibodies isolated from VITT patients with the coronavirus spike protein (and only a few out of 200 patients recovered from COVID-19 reacting weakly with PF4) [104], they now show in a recent preprint close proximity between PF4 molecule, platelets, and adenovirus hexon capsid protein-positive staining particles. This association in the complex could be displaced by high concentrations of (highly negatively charged) heparin. Furthermore, cross reactivity of endogenous immunoglobulins in the serum of healthy, non-vaccinated volunteers with the vaccine proteins was found which, however, was substantially stronger in sera from VITT patients. Importantly, the incubation of purified neutrophils with PF4 and serum from VITT patients triggered the formation of procoagulant NETs. Similarly, the respective affinity-purified anto-PF4 antibodies activated neutrophils. PF4/VITT serum stimulated NET formation was strongly enhanced in the presence of platelets, while sera from healthy controls were inactive [105].

An alternative charge-based hypothesis has been proposed. Among the 50 billion virus particles in each dose, some may break apart and release their DNA. Similarly to heparin, DNA is negatively charged and can bind to the positively charged PF4. The complex might then trigger the production of antibodies, especially when the immune system is already on high alert because of the vaccine. An immune reaction to extracellular DNA is part of an ancient immune defense triggered by severe infection or injury, and free DNA itself can signal the body to increase blood coagulation [7,106].

### Additional Activities of PF4

Activated platelets release micromolar concentrations of the chemokine CXCL4/platelet factor-4. Deposition of CXCL4 onto the vascular endothelium is involved in atherosclerosis, facilitating monocyte arrest and recruitment. CXCL4 was shown to drive chemotaxis of the monocytic cell line THP-1 and induced CCR1 endocytosis and monocyte chemotaxis in a CCR1 antagonist-sensitive manner [107]. The role of the platelet chemokine platelet factor 4 (PF4) in hemostasis and thrombosis has been described [108].

## 9. The Impact of the Adenoviral Vector

One important factor contributing to the thrombotic events observed following vaccination with adenoviral vector-based anti SARS-CoV-2 vaccines may rely on the adenoviral vector used to deliver the transgene. The recombinant adenovirus ChAdOx1 used in the ChAdOx1 nCoV-19 vaccine against SARS-CoV-2 is replication defective in normal cells. Adenovirus genes of 28 kbp are delivered to the cell nucleus alongside the SARS-CoV-2 S glycoprotein gene. A recent study used direct RNA sequencing to analyse transcript expression from the ChAdOx1 nCoV-19 genome in human MRC-5 and A549 cell lines that are non-permissive for vector replication alongside the replication permissive cell line HEK293. The expected SARS-CoV-2 S coding transcript dominated in all cell lines. However, rare S transcripts with aberrant splice patterns or polyadenylation site usage were also found. Adenovirus vector transcripts were almost absent in MRC-5 cells, but in A549 cells a broader repertoire of adenoviral gene expression at very low levels was observed. In addition to S glycoprotein, multiple adenovirus proteins were detected in A549 cells. The study demonstrated that low levels of viral backbone gene transcription take place alongside very high levels of SARS-CoV-2 S glycoprotein gene transcription following SARS-CoV-2 vaccine ChAdOx1 nCoV-19 infection of human cell lines [109].

Adenoviral vectors are highly immunogenic and elicit robust transgene antigen-specific cellular and humoral immune responses. Furthermore, the excessive induction of Type I interferons by some adenoviral vectors has been shown. Entry factor binding or receptor usage of distinct adenoviral vectors can affect their in vivo tropism following administration by different routes [110].

Intravenous (i.v.) delivery of recombinant adenovirus serotype 5 (Ad5) vectors for gene therapy is hindered by safety and efficacy problems. Most i.v.-delivered Ad5s are sequestered in the liver, and animal studies indicate that Kupffer cells (KCs) play a major role in this trapping. Studies have shown that liver sequestration is not mediated by the Ad5 receptor CAR but involves either a direct or a blood factor (coagulation factors IX and X and complement protein C4BP) mediated interaction between the Ad fiber and cellular proteoglycans. After i.v. administration, Ad5 rapidly binds to circulating platelets, which causes their activation/aggregation and subsequent entrapment in liver sinusoids. Virus-platelet aggregates are taken up by Kupffer cells and degraded. Adenovirus sequestration in organs could be reduced by platelet depletion prior to vector injection [111].

Importantly, thrombocytopenia has been consistently reported following the administration of adenoviral gene transfer vectors. In one study, the authors assessed the influence of von Willebrand Factor (VWF) and P-selectin on the clearance of platelets following adenovirus administration. In mice, thrombocytopenia occurs between 5 and 24 h after adenovirus delivery. The virus activated platelets and induced platelet-leukocyte aggregate formation. Adenovirus-induced endothelial cell activation was shown by VCAM-1 expression on virus-treated, cultured endothelial cells and by the release of ultra large molecular weight multimers of VWF within 1 to 2 h of virus administration with an accompanying elevation of endothelial microparticles. In contrast, VWF knockout (KO) mice did not show significant thrombocytopenia after adenovirus administration. Furthermore, it was shown that adenovirus interferes with the adhesion of platelets to a fibronectin-coated surface, and flow cytometry revealed the presence of the Coxsackie adenovirus receptor on the platelet surface. They concluded that VWF and P-selectin are critically involved in a complex platelet–leukocyte–endothelial interplay, resulting in platelet activation and accelerated platelet clearance following adenovirus administration [112].

Furthermore, replication-deficient adenoviruses are known to induce acute injury and inflammation of infected tissues, thus, limiting their use for human gene therapy. Chemokine expression was evaluated in mice following intravenous administration of various adenoviral vectors. Administration of adenoviral vectors coding for beta-galactosidase (adCMVβgal), or for Green Fluorescence Protein (adCMV-GFP), or FG140 intravenously rapidly induced a dose-dependent expression of C-X-C and C-C chemokines in the liver. Infection with adCMVβgal resulted in rapidly increased hepatic levels of MIP-2 mRNA and MCP-1 and IP-10 mRNA levels which peaked at 6 h with >25-fold and >100-fold expression, respectively. Additionally, RANTES and MIP-1beta mRNA were induced. The induction of chemokines occurred independently of viral gene expression since psoralen-inactivated adenoviral particles produced an identical pattern of chemokine gene transcription. The expression of chemokines correlated with the influx of neutrophils and CD11b+ cells into the livers. At high titers, all adenoviral vectors caused significant hepatic necrosis and apoptosis following systemic administration. Pretreatment of animals with neutralizing anti-MIP-2 antibodies or neutrophil depletion reduced serum ALT/AST levels and attenuated adenovirus-induced hepatic injury, indicating that liver injury following adenoviral vector application is largely due to chemokine production and neutrophil recruitment [113].

Moreover, the gene transfer to the respiratory tract by replication-deficient adenoviruses is limited by the induction of inflammatory and immune responses. E1-E3-deleted recombinant adenoviruses carrying the cystic fibrosis gene (Ad.CFTR) were shown to increase the expression of proinflammatory intercellular adhesion molecule-1 (ICAM-1), together with enhanced translocation of NF-κB into the nucleus and enhanced binding to the NF-κB consensus sequence on the ICAM-1 promoter in respiratory epithelial A549. The CFTR-dependent increase in ICAM-1 mRNA was abolished by inhibitors of NF-κB which abolished both Ad.CFTR-induced NF-κB DNA binding and transactivating activities. These results indicate a critical role of NF-κB in proinflammatory responses induced by replication-deficient adenoviral vectors in respiratory cells [114].

It was tested whether the early phase of the virus–cell interaction is sufficient to stimulate ICAM-1 upregulation. A549 cells were infected with wild-type Ad5 (Ad5) or adenoviral vector coding for the cystic fibrosis gene (Ad.CFTR) or to Ad5 inactivated by heating up to 56 °C (Ad5/56 °C). All Ad5, Ad.CFTR, and Ad5/56 °C activated NF-κB and increased ICAM-1 mRNA levels within 4 h after exposure. The role of different signal transduction pathways in ICAM-1 mRNA induction was studied. ICAM-1 mRNA upregulation was inhibited by ERK1/2 inhibitors (>70% inhibition), inhibitors of the p38/MAPK pathway (50% inhibition), of the JNK pathway (>80% inhibition), and of the NF-κB pathway (>95% inhibition). The data indicate a link between the activation of the three major MAPK pathways, NF-κB, and the upregulation of ICAM-1 gene expression induced by adenoviral vectors in the initial phase of infection [115]. Adenovirus serotypes 2 or 5 (Ad2/5) enter respiratory epithelia after initial binding of fiber with the coxsackie-adenovirus receptor (CAR) or, alternatively, with cell surface heparan sulfate glycosaminoglycans. Ad2/5 internalization is triggered by the binding of penton bases to cellular RGD-binding inte-grins. The interaction of Ad fibers with CAR activated all ERK1/2 and JNK MAPK and the nuclear translocation of NF-κB. Moreover, the interaction of Ad fibers with CAR promoted transcription of different chemokines including IL-8, GRO-alpha, GRO-gamma, RANTES, and IP-10. These results indicate the binding of Ad5 fibers with cellular CAR as a key proinflammatory activation event that is independent of the transcription of Ad5 genes [116].

Another study tested a first-generation recombinant adenoviral empty vector, rAdE1, regarding its adjuvant effect independently of its vector function. Mice received one injection of a mixture of six lipopeptides (LP6) used as a model immunogen, together with AdE1. When coinjected with a suboptimal dose of lipopeptides, AdE1 significantly increased the effectiveness of the vaccination. In contrast to mice that received LP6 alone or LP6 plus a mock adjuvant, mice injected with AdE1 plus LP6 developed both a polyspecific T-helper type 1 response and an effector CD8 T-cell response. When mice were immunized with LP6 and each individual capsid component, i.e., hexon, penton base, or fiber, the hexon protein was found to be responsible for the adjuvant effect [117].

Recombinant adenoviral vectors (rAd) have been employed to transduce dendritic cells (DCs). Following rAd infection, mouse bone marrow-derived immature DCs were shown to upregulate the expression of MHC class I and II antigens, costimulatory molecules (CD40, CD80, and CD86), and the adhesion molecule ICAM-1. rAd-transduced DCs exhibited increased allostimulatory capacity and induced the expression of IL-6, IL-12p40, IL-15, gamma interferon, and TNFα mRNAs. These effects were not related to specific transgenic sequences or to rAd genome transcription. The rAd effect correlated with a rapid increase in NF-κB DNA binding activity. Moreover, adenoviral vector-induced DC maturation was inhibited by the proteasome inhibitors or by infection with rAd-IκB, a rAd-encoding the dominant-negative form of IκB. After i.v. administration, adenoviral vectors were rapidly entrapped in the spleen by marginal zone DCs, suggesting that rAd also induces DC differentiation in vivo. These data indicate additional mechanisms for the high immunogenicity of adenoviral vectors [118].

Altogether, these data indicate that even E1–E3 deleted recombinant adenoviral vectors such as those used in ChAdOx1 nCov-19 (AZD1222; AstraZeneca) and Ad26.COV2.S (Johnson & Johnson/Janssen) adenoviral vector-based vaccines are powerful activators of the immune system, particularly for stimulating the innate immune system at various stages. Various hypotheses about the vaccine components that could be involved, including adenovirus-derived substances, have been raised [105]. Adenoviral vectors can bind to circulating platelets, which causes their activation and aggregation and subsequent entrapment in liver sinusoids [111,112,119]. Generally, the activation of platelets is known to result in increased release of PF4. As adenoviruses are known to activate platelets, it would be plausible that the replication-deficient adenoviral vector could be directly responsible for the release of platelet-derived PF4, which as highly positively charged PF4 tetramers can bind to negatively charged glycosaminoglycan structures on endothelial cells [120]. This implies the release of significant amounts of vaccine particles into the bloodstream, which is what would happen in cases of inappropriate administration, with damage or injection into small blood vessels. However, it is unknown how frequently this may happen after intramuscular injection.

There are alternative scenarios under discussion that involve endothelial cells as a central cellular targets [105,120]. Indeed, endothelial cells are efficiently transduced upon intramuscular injection [121]. Various studies in vitro and in vivo suggest that transduced endothelial cells might be directly damaged by the spike protein that they synthesize [122,123]. Furthermore, endothelial cells might expose the spike protein on their luminal side, e.g., bound to proteoglycans as heparan sulfate containing proteoglycans were shown to serve as attachment factors for the spike protein [124]. Platelets might then be recruited and activated by the spike protein bound to endothelial cells [125]. Furthermore, PF4 released by activated platelets could combine with anionic proteoglycans shed from endothelial cells. In such a scenario, both the adenovirus and the spike protein would contribute to the formation of immunogenic PF4 following vaccination with adenoviral vector-based SARS-CoV-2 vaccines [120].

## 10. Additive and Synergistic Effects of the S Protein and Adenoviral Vector

### 10.1. Antigen Mimickry

A recent publication has presented data that indicate that the severe side effects observed in rare cases may have to be attributed to adenoviral vaccines. Transcription of wild-type and codon-optimized spike open reading frames enables alternative splice events that result in C-terminal truncated, soluble spike protein variants. These soluble spike variants may initiate severe side effects when binding to ACE2-expressing endothelial cells in blood vessels. Analogous to the thromboembolic events caused by the spike protein encoded by the SARS-CoV-2 virus, the underlying disease mechanism has been termed “Vaccine-Induced COVID-19 Mimicry” syndrome (VIC19M syndrome) vector-based vaccines [126].

### 10.2. NF-κB Promotes Leaky Expression of Adenovirus Genes in a Replication Incompetent Adenovirus Vector

Systemic administration of Ad vectors was found in certain cases to result in severe hepatotoxicities, which may partly be due to the leaky expression of Ad genes in the liver. It was shown that NF-κB mediates the leaky expression of Ad genes from the Ad vector and that the inhibition of NF-κB can suppress Ad gene expression and hepatotoxicities following application of Ad vectors. Activation of NF-κB by recombinant TNFα significantly enhanced the leaky expression of Ad genes, whereas Ad gene expression was suppressed by inhibitors of NF-κB or siRNA-mediated knockdown of NF-κB. An adenoviral vector encoding for the dominant-negative IκBα (Adv-CADNIκBα) mediated 70% suppression of the leaky expression of Ad genes in the liver. Importantly, Adv-CADNIκBα did not induce apparent hepatotoxicities. These results indicate that the inhibition of NF-κB results in the suppression of adenoviral vector-mediated tissue damage by both suppression of inflammatory responses and reduction in the leaky expression of adenoviral genes [127].

Overall, these studies referenced in the present and previous chapters demonstrate that both SARS-CoV-2 spike protein AND adenoviral vectors activate the NF-κB pathway together with several additional signal transduction pathways, which can result in additive and synergistic effects

## 11. Adenoviral and Producer Cell Protein Impurities, and Additives in the ChAdOx1 nCov-19 Vaccine Formulation

Several ChAdOx1 nCoV-19 vaccines have been analyzed by biochemical and proteomic methods. A recent preprint has shown that the vaccine, in addition to the adenovirus vector, contained substantial amounts of both human and nonstructural viral proteins. The authors analyzed three different lots of the ChAdOx1 nCov-19 vaccine by SDS polyacrylamide gel electrophoresis (SDS-PAGE) followed by silver staining and compared the staining pattern of the separated proteins with those of HAdV-C5-EGFP, an adenovirus vector purified by CsCl ultracentrifugation followed by mass spectrometry. Based on intensity comparisons of LC/MS signals, they estimated that in one of three lots, approximately 2/3 of the detected protein amounts were of human and 1/3 of virus origin, while the two other lots consisted of rather equal amounts of human and viral proteins. In addition to the expected viral proteins that form the virion (hexon, penton base, IIIa, fiber, V, VI, VII, VIII, IX, and others), several nonstructural viral proteins were also detected at high abundancy, although they are not part of the mature viral particle. Furthermore, peptides from more than 1000 different human proteins derived from the human vector production cell line were detected. The detected proteins were derived from different cellular compartments, including the cytoplasm, nucleus, endoplasmic reticulum, Golgi apparatus, and others. Intriguingly, among the human proteins found in the vaccine and in addition to several cytoskeletal proteins, including Vimentin, Tubulin, Actin, and Actinin, the group of heat shock proteins (HSPs) and chaperones stood out in abundancy. Among the most abundant proteins (including viral proteins), HSP 90-beta and HSP-90-alpha were cytosolic HSPs (with 9.5% and 4.3% of the total protein), and three chaperones of the endoplasmic reticulum (transitional endoplasmic reticulum ATPase, endoplasmin, and calreticulin) were present. Extracellular HSPs are known to modulate innate and adaptive immune responses; they can exacerbate pre-existing inflammatory conditions, have been associated with autoimmunity, and can even become targets of autoimmune responses themselves. They very efficiently initiate specific immune responses by receptor-mediated uptake of HSP-peptide complexes in antigen-presenting cells (APCs), mainly via CD91 and scavenger receptors. Furthermore, among the viral proteins detected, the adenoviral penton base is another candidate for inducing early toxicity via an RGD motive present in a solvent-exposed loop of penton base by interacting with integrins on cell membranes, including platelets [128].

Similarly, a recent preprint showed that, in addition to the main SARS-CoV-2 spike protein, multiple additional known and unknown proteins including vector proteins, human proteins, including human membrane proteins, and various additives such as sucrose, EDTA, and histidine are present. Relevant amounts of EDTA (0.0002 mM) were found to induce platelet activation in whole blood of healthy volunteers as assessed by flow cytometry. Furthermore, intradermally injected ChAdOx1 nCoV-19 vaccine increased vascular leakage in a Miles skin edema mouse model. Edema formation was likely due to EDTA in the carrier solution as EDTA alone triggered edema and vascular leakage to a similar extent as the vaccine [105]

## 12. Discussion of an Integrated Mechanistic Model for VITT Following Vaccination with Adenovirus Vector-Based Vaccines

SARS-CoV-2 spike protein is known to bind to the ACE2 receptor followed by endocytosis [13], whereas the primary mode of adenovirus (including replication defective E1/E3 deletion variants) is the high affinity binding of the fiber-knob to the CAR (coxsackie and adenovirus receptor), followed by interaction between the arginine-glycine-aspartate (RGD) motif within the viral penton base with the integrins on the cell surface, which facilitates viral internalization [110] (see Figure 1). In addition to the primary high affinity binding sites for cellular uptake, both SARS-CoV-2 and adenovirus have been described to bind and activate various Toll-like receptors (TLRs) [77,110,129,130]. All these binding events of both SARS-CoV-2 S-protein and adenoviral vector have been described to activate—among various other—the NF-κB signaling pathway by activating IκB kinase (IKK) complexes, release of p50/p65 heterocomplex from the inhibitor complex with IκBa by proteasomal degradation of IκB, and translocation of the released and phosphorylated p50/p65 heterocomplex to the nucleus where the transcription of a huge panel of pro-inflammatory genes, including those for cytokines, such as TNFα, IL-1, and IL-6; chemokines, such as MCP-1 and MCP-3; adhesion molecule, such as ICAM-1 and VCAM-1; and complement components and coagulation factors, such as PAI-1 [84], are induced. These processes are triggered in a broad variety of cell types that express the respective receptors, including epithelial cells (e.g., in the lung), macrophages (including alveolar macrophages), and endothelial cells, found in all organs [77,78,79,80]. Activations of additional signal transduction pathways, such as JAK/STAT3 pathway, are activated by the NF-κB triggered release of IL-6, colony stimulating factors, and various G-protein binding chemokines [83,88,89,90,91].

Expression of proinflammatory cytokines such as TNFα, IL-1, and IL-6 together with increased integrin expression [131,132] can result in autoamplifying positive feedback loops [33,34] and also affect other integrated systems such as complement and coagulation [39,40,41,42,43,44]. The release of various chemokines, particularly MCP-1 and IL-8, stimulates the migration and accumulation of various inflammatory and immune cells at the site. In particular, neutrophils are attracted, activated by the proinflammatory cytokine/chemokine milieu, and stimulated to interact with endothelial cells and platelets. The release of DNA chromatin nets, i.e., NETosis, has been described and is considered as one major mechanism for thromboses and occlusion of capillaries in COVID-19 [45,46,47,48].

While these events are known to accelerate and amplify in the case of massive viral infection by SARS-CoV-2 resulting in COVID-19, the localized expression of the spike protein in case of vaccination is not expected to result in levels resulting in a major systemic symptomatic picture. However, in the case of adenoviral vector vaccines, additive and, possibly in some cases, synergistic effects of the adenoviral vector itself were observed. Moreover, in an additional activation of the NF-κB pathway—which may be considered a vaccine adjuvant effect—adenoviruses have the ability to bind and activate platelets and endothelial cells [111,112,119]. Adenoviral vectors can bind to circulating platelets, which causes their activation and aggregation and subsequent entrapment in liver sinusoids. Moreover, the activation of platelets is known to result in increased release of PF4—a tetramer with high positive surface charge, which on the one hand can bind to negatively charged glycosaminoglycan structures on endothelial cells. As adenoviruses are known to activate platelets, it is plausible that the replication-deficient adenoviral vector could be directly responsible for the release of platelet-derived PF4. However, this hypothesis implies that significant amounts of vaccine particles would reach the bloodstream after intramuscular injection, which seems less likely. An alternative scenario would involve endothelial cells. Indeed, endothelial cells are efficiently transduced upon intramuscular injection. Transduced endothelial cells might be directly damaged by the spike protein that they synthesize. Platelets might then be recruited and activated by the spike protein expressed by endothelial cells. PF4 released by activated platelets may combine with anionic proteoglycans located on the surface of endothelial cells or be shed from endothelial cells. In such a scenario, both the adenoviral vector and the spike protein would contribute to the formation of immunogenic PF4 following vaccination with adenoviral vector-based COVID-19 vaccines [50,120].

One important central point of the VITT is the anti-PF4 antibodies found in the majority of patients with venous thromboses following SARS-CoV-2 adenoviral vector-based vaccines. The VITT-induced antibodies are most likely induced by electrostatic binding of the positively charged PF4 tetramers relative to the negative surface of the adenoviral vector hexones. In this complex, conformational changes occur—similar to those reported for heparin-induced HIT—which makes them highly immunogenic, triggering the formation of anti-PF4-directed antibodies. ChAdOx1 viral capsid has been shown to have the highest electronegative potential compared to other adenoviruses tested due to differences in the ChAdOx1 hexon hypervariable regions (HVRs) compared to other adenoviruses. This particularly high electronegative surface potential of the ChAdOv1 viral capsid is expected to increase the strength of incidental charge-based interactions with other molecules, such as PF4 [103]. In contrast, the Ad26 adenovirus used in the J&J COVID-19 vaccine exposes a lower surface charge, which may explain why VITT seems to be less commonly observed in recipients of the J&J vaccine. Moreover, the Ad26 and Ad5 adenoviruses used in the Sputnik V COVID-19 vaccine have lower negative surface charges compared to ChAdOx1 correlating with no VITT reports so far [133].

The binding of these induced antibodies to the PF4-attached adenoviral surface will facilitate uptake of these complexes via an FcgammaR-mediated process. While macropages, monocytes, NK cells, and DCs will take up antibody-PF4 (adenovirus) complexes via their different FC gamma receptors, resulting in the activation of the cells, Fc gammaR-induced complement activation, and degradation and sequestration of adenoviral vectors in reticulendothelial cells. Platelets are also able to bind such complexes via their FcgammRIIA receptor, resulting in the activation and release of PF4, aggregation of platelets, and resulting thromboses [120].

There is an additional mechanism that might accelerate the induction of anti-PF4 antibodies, which is the supposed superantigen feature of the spike protein. Different from antigen-specific activation of single T-cell clones in the case of conventional antigens, with the antigen presented in the MHC-TCR groove, superantigens have been described to bind outside this groove to the MHC and/or TCR, resulting in polyclonal activation [64,65,66]. This, together with the highly immunogenic PF4-adenovirus complexes, may facilitate the induction of PF4-specific antibodies. Notably, both normal cell activation and superantigen-induced polyclonal T cell activations are dependent on NFκB pathway signaling [72,73,74,75,76].

Although originally being locally applied, a limited local dissemination of adenoviral vaccines, e.g., via binding to endothelial cells [134,135,136] together with the described NF-κB-triggered leaky expression of adenovirus genes in originally replication-incompetent adenoviral vectors [127] may result in rare cases to self-amplifying cascades resulting in activated or damaged endothelial cells, activated and aggregated platelets, and activation of the coagulation system at sites distant to the application site, i.e., systemic prothrombotic procoagulation events together with a corresponding (consumption) thrombocytopenia as observed in rare cases of adenovirus vector-based SARS-CoV-2 vaccines [39,40,41,42,43,44].

Additional activation of the described molecular and cellular pathways may occur by impurities, such as the significant amounts of human heat shock proteins found in several ADT1222 vaccine preparations [105,128].

Together, these molecular and cellular mechanisms described in the previous sections may explain the higher prevalence of rare thromboses and thrombocytopenia following adenoviral vector-based anti-SARS-CoV-2 vaccines compared to mRNA-based anti-SARS-CoV-2 vaccines.

## Figures and Tables

**Figure 1 ijms-22-10791-f001:**
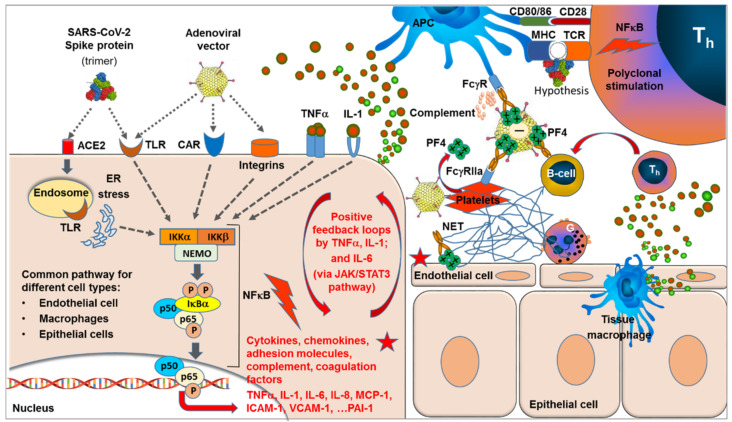
SARS-CoV-2 spike protein (expressed following vaccination with either mRNA or adenoviral vector) binds to the ACE2 receptor followed by endocytosis. Binding of the adenovirus (including replication defective E1/E3 deletion variants) occurs via binding of the fiber-knob to the CAR (coxsackie and adenovirus receptor), followed by interaction between the arginine-glycine-aspartate (RGD) motif within the viral penton base with the integrins on the cell surface. In addition both, SARS-CoV-2 and adenovirus bind and activate various Toll-like receptors (TLRs), leading to activation of the NFκB signaling pathway, including activation of IKK complexes, release of p50/p65 complexes from the inhibitor complex with IκBa, and translocation of the phosphorylated p50/p65 heterocomplex to the nucleus, where the transcription of proinflammatory genes, including those for cytokines, such as TNFα, IL-1, and IL-6; chemokines, such as MCP-1 and MCP-3; adhesion molecules, such as ICAM-1 and VCAM-1; and complement components and coagulation factors, such as PAI-1, is induced. These processes are triggered in a broad variety of cell types that express the respective receptors, including epithelial cells, macrophages, and endothelial cells. Expression of TNFα and IL-1 and increased integrin expression can result in auto amplifying positive feedback loops, which may extend to other integrated systems such as complement and coagulation. Signaling of IL-6 via JAK/STAT3 pathway is expected to trigger additional amplifying loops. The release of chemokines, such as MCP-1 and IL-8, will attract the migration and accumulation of inflammatory and immune cells to the site. In particular, neutrophils are attracted, releasing DNA chromatin nets, i.e., NETs. While these events are known to accelerate and amplify in the case of massive viral infection by SARS-CoV-2, the localized expression of the spike protein in the case of vaccination was not expected to result in levels inducing a major systemic symptomatic picture. However, in the case of adenoviral vector vaccines, additive and synergistic effects are possible. In addition to additional activation of the NFκB pathway, adenoviruses directly bind and activate platelets, which causes their activation/aggregation. Activation of platelets results in increased release of PF4—a tetramer with high positive surface charge—which can bind to the negatively charged glycosaminoglycan on endothelial cells. VITT-induced antibodies are induced by electrostatic binding of the positively charged PF4 tetramers to the negative surface of the adenoviral vector hexones forming highly immunogenic complexes, which trigger the formation of anti-PF4-directed antibodies. Binding of induced antibodies to the PF4-attached adenoviral surface facilitates uptake of these complexes via an Fc-gammaR-mediated process. Antigen presenting cells (APC) take up antibody-PF4 (adenovirus) complexes via different types of Fc-gamma receptors, resulting in cellular activation and complement activation. Additionally, platelets are able to bind such complexes via their FcgammaR2a receptor, resulting in further activation, release of PF4, aggregation of platelets, and thromboses. The supposed superantigen features of the spike protein may present an additional mechanism for accelerating the induction of anti-PF4 antibodies. Different from antigen-specific activation of single T-cell clones in the case of conventional antigens, with the antigen presented in the MHC-TCR groove, superantigens bind outside this groove to the MHC and/or TCR, resulting in polyclonal activation. Adapted from ref. [38].

## Data Availability

Not applicable.

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
