# Peer review of "Coagulopathies after Vaccination against SARS-CoV-2 May Be Derived from a Combined Effect of SARS-CoV-2 Spike Protein and Adenovirus Vector-Triggered Signaling Pathways"

_ijms, 2021, doi:10.3390/ijms221910791_

Round 1

Reviewer 1 Report

This is a well written review of a potentially interesting topic, and I gather it is part of a special issue of related reviews.  On the positive side, there may be interest in the likely significant biological relevance of this coagulation side effect after vaccination with Adenovirus Vector-based vaccine against SARS-CoV-2. As described in the specific comments below, I think there are a few aspects of this topic that should also be discussed to give the reader a better idea of the complexities that are glossed over in the figures.  On the negative side, while the topic is interesting, the molecular mechanisms of the discussed mechanisms are ill-defined, and there has been little progress in till now.  I would encourage author to explain these issue also as listed below.

  1. There are several other possibilities including the entry of the Adenovirus (Ads) into the bloodstream, either through inappropriate vaccine administration or other mechanisms. this hypothesis could be quite convincing as Greinacher et al, found that the Oxford/AstraZeneca vaccine could bind with PF4 and induce pro-inflammatory immune responses at the site of injection.
  2. Other studies talk about the protein-based impurities in the vaccine formulation because of the different purification process may causing these sided effects.
  3. If possible to discuss, I would encourage authors insight what would be the possible solution to overcome these side effect. Different route of administration or potentially new vectors would might help to overcome this problem.

Author Response

Thank you for the recommendations. All raised points have been addressed in the edited manuscript, all changes are tracked (track changes modus) and most important points are marked colored.

Ad 1. additional possibilites such as entry of adenovirus into bloodsteam and additional mechanisms, such as transduction and activation or damage of endothelial cells are discussed (yellow marked text) in sections 9 and 12, including addional references.

ad 2. studies about protein-based impurities in vaccine formulations are cited and discussed (yellow marked text section 11, inclusing additional references)

ad 3. possible solution to overcome /lower the side effects of adenoviral vectors for COVID-19 vaccines (yellow marked text in section 12)

I hope that the modifications and additions to the text make the manuscript suitable for publication in IJMS

Reviewer 2 Report

The manuscript by Ralf Kircheis is well written and touches on a very acute topic at this moment. There are some recommendations which author has to consider before the paper will be published. 

  1. Since phosphorylation of the canonic STAT3 pathway is very important in both COVID-19-induced acute lung injury, systemic inflammation and also and coagulopathy pathogenesis, the author should to add a small paragraph about STAT3 pathway.
  2. The author should pay more attention to animal studies about COVID-19. The discussion about NFkB pathway in the mouse model of SARS-CoV-2 Spike protein-induced acute respiratory distress syndrome (Colunga Biancatelli etc., The SARS-CoV-2 spike protein subunit S1 induces COVID-19-like acute lung injury in K18-hACE2 transgenic mice and barrier dysfunction in human endothelial cells) should be added to paragraph 6.3.

Author Response

Thank you very much for your recommendations. All points raised by the reviewer have been addressed in the modified manuscript, all changes are marked by track changes modus, most important points are marked by color.

ad. 1 An additional sub chapter addressing the involvement of JAK/STAT3 pathway has been included (new section 7), relevant references have been included, Figure 1 has beed modified with reference to the positive feedback loops by JAK/STAT3 pathway

ad. 2 animal models for studying mechanisms of COVID-19 have been included, with reference to the study by Colunga Biancatelli et al., as part of section 6.2.

Additionally, previous section 10 (NF-kappaB activation is central to coagulation events) was shifted to section 6 (now being sub part 6.3)

I hope that the modifications to the manuscript make manuscript suitable for publication in IJMS